# The Impact of Uncertainty in Macroeconomic Variables on Stock Returns in the USA

Leonardo Iania [1,*] , Robbe Collage [2] and Michiel Vereycken [2,*]

[1] LIDAM/CORE/LFIN, UCLouvain, Voie du Roman Pays 34, 1348 Louvain la Neuve, Belgium
[2] Faculty of Economics and Business, KU Leuven, Naamsestraat 69, 3000 Leuven, Belgium
* Correspondence: leonardo.iania@uclouvain.be (L.I.); michiel.vereycken@outlook.be (M.V.)

**Abstract:** In this research paper, the impact of macroeconomic uncertainty on stock returns in the United States of America is examined. To measure this macroeconomic uncertainty, a survey of Consensus Economics with data ranging from 1989 until 2019 was employed. The survey consists of monthly forecasts for several macroeconomic variables for multiple countries. Four uncertainty measures were developed, based on the standard deviation, interquartile range, high-minus-low and an AR- and GARCH model. By performing linear regressions, a positive relationship between macroeconomic uncertainty and stock returns was identified for, on average, 13 out of 49 sectors, which is consistent with economic theory. Furthermore, the standard deviation of stock returns was regressed on macroeconomic uncertainty. A positive relationship was found for, on average, 41.7 out of 49 sectors. The results are discussed at a general level, at the level of the macroeconomic variables and at the sector level.

**Keywords:** stock returns; uncertainty; risk factors; surveys

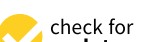



## 1. Introduction

Macroeconomic variables play an important role in financial markets as they determine the state of the economy. Following the efficient market hypothesis, all information should be incorporated into prices (Fama (1970)). Hence, a change in these variables expands the information set of traders and should therefore automatically reflect changes in stock prices and, consequently, stock returns. This is because of a change in traders' beliefs about the fundamental value of the stock after the news has arrived.

Extensive research concerning the relationship between macroeconomic variables and stock returns can be found (see Chen et al. (1986), Humpe and Macmillan (2009), Asprem (1989) and many more). On the contrary, the literature on the impact of uncertainty in these macroeconomic variables on financial markets is rather limited. However, since the COVID-19 pandemic starting in March 2020, the economy has been dominated by uncertainty. At the start of the pandemic, no economist could anticipate how bad the economy would suffer from the massive lockdowns. Afterwards, inflation started rising caused by supply chain disruptions. Uncertainty peaked when Russia invaded Ukraine, which led to an energy crisis and inflation skyrocketing. Central banks considered inflation as temporary for too long, leading them to increase interest rates at a massive speed. In the meantime, China was confronted with a real estate sector on the brink of collapse and cities in lockdown because of a surge in COVID-19 cases. These events introduced a high degree of uncertainty in macroeconomic variables such as GDP growth, inflation, consumer spending etc., and had a severe impact on financial markets. Linking back to the efficient market hypothesis, this study examined how uncertainty in a trader's information set is incorporated into prices.

The goal of this paper is to provide an empirical assessment of the impact of uncertainty on stock market returns. We proposed survey-based measures of uncertainty

and investigated to what extent they explain stock returns, next to the well known Fama–French factors.

The results are discussed at three different levels: at a general level, at the level of the macroeconomic variables and at the sector level. At a general level, a significant impact of uncertainty in macroeconomic variables on stock returns is identified for 13 out of 49 industries on average. Regarding the impact on the standard deviation of returns, a significant effect was found for on average 41.7 out of 49 industries.

Our contribution to the literature is twofold. First, we introduce new measures of uncertainty, based on a survey of professional forecasters. Second, we show that our measures of uncertainty significantly effect stock returns in the USA and decompose this general effect into the impact on 49 different industries.

The structure of the paper will be as follows. First, an overview of the existing literature is provided. Next, the data sets used to pursue this research are described, with additional information concerning the specified dependent and independent variables. Third, the methodology section will elaborate on the different econometric models and regressions executed to obtain the results. The findings are then discussed in the results section. To conclude, the main findings will be summarised and an overview of the implications of our findings will be provided.

## 2. Literature Review

### 2.1. Macroeconomic Variables and Stock Returns

The relation between macroeconomic variables and stock returns has been well-studied. Chen et al. (1986) defines state variables that describe the economy and investigate their impact on asset prices. Asset prices are determined by discounting future cash flows at the appropriate discount rate. They examine which state variables impact these two factors and found industrial production, changes in risk premia, and twists in the yield curve to be significant determinants of asset prices. By contrast, changes in expected inflation only weakly explained asset prices, while consumption and oil prices were found to be insignificant state variables.

Asprem (1989) conducted a study concerning the effect of macroeconomic variables on stock prices for ten European countries. It was shown that employment, imports, inflation, and interest rates are negatively related to changes in stock prices, while future real activity, measures for money, and the U.S. yield curve are positively related. Moreover, real interest rate changes and oil price changes significantly impact stock returns in Norway (Gjerde and Saettem (1999)). Cheung and Ng (1998) used quarterly data of Canada, Germany, Italy, Japan, and the U.S. and found the real oil price, real output, real money supply, and real consumption to be significant state variables in explaining national stock market indices.

Focusing on the United States, Humpe and Macmillan (2009) found stock prices to be positively related to industrial production, while a negative relationship was identified for the consumer price index and the long term interest rate. The relationship between stock returns and the money supply was found to be insignificant. Ratanapakorn and Sharma (2007) confirms the negative relation between stock prices and the long-term interest rate and the positive relation between stock prices and industrial production. Additionally, they report a positive relation between stock prices and the money supply, inflation, the exchange rate and the short-term interest rate.

### 2.2. Uncertainty in Macroeconomic Variables

Jurado et al. (2015) questioned whether commonly used proxies of uncertainty (e.g., the appearance of certain "uncertainty related" key words in news publications or the cross-sectional dispersion of survey-based forecasts) truly reflect economic uncertainty. They found a negative answer to this question and introduce their own measure of uncertainty, defined as "the common volatility in the unforecastable component of a large number of economic indicators". With this uncertainty measure, they identify fewer but more severe uncertainty periods compared with common proxies of uncertainty.

Rossi and Sekhposyan (2015) started from the insights of Jurado et al. (2015) and introduced a more general uncertainty measure, in which they relied on the unconditional likelihood of the observed outcome. Furthermore, Segal et al. (2015) make a distinction between good and bad uncertainty. They define good and bad uncertainty as the variance associated with the respective positive and negative innovations in an underlying macroeconomic variable. In their paper, it is shown that good and bad macroeconomic uncertainty have a significant and opposite impact on growth and asset valuations.

Zooming in on survey-based measures, Chuliá et al. (2017) concluded that the dispersion in the outcome of these surveys is more an expression of different forecaster opinions than of actual uncertainty. In the paper of Lahiri and Sheng (2010), this is investigated in more detail. Cascaldi-Garcia et al. (2020) argue that only surveys that ask about the uncertainty of the forecasts, by for example probabilistic responses, are an appropriate measure of uncertainty. Such research has been conducted by D'Amico and Orphanides (2008). Jurado et al. (2015) list a number of drawbacks for using analysts' forecasts to measure uncertainty. A limited number of series, biased forecast(er)s, and the dispersion in forecasts being due to differences in opinion rather than to uncertainty, are some of these drawbacks.

### 2.3. Uncertainty in Macroeconomic Variables and Stock Returns

While the risk-return trade-off, introduced by Markowitz (1952) and formalised by Sharpe (1964) and Lintner (1965), is well known in financial academic literature, Anderson et al. (2009) introduced another trade-off, namely the uncertainty-return trade-off for which more significant empirical evidence is found. They define uncertainty as the degree of disagreement between professional forecasters. In contrast to parts of the literature that focus on firm-specific measures of uncertainty (Anderson et al. (2005), Diether et al. (2002)), Anderson et al. (2009) used aggregate measures of disagreement. However, they attribute different weights across forecasters and find that disagreement only matters with unequal weighting schemes. Also Kogan and Wang (2002) find that a positive uncertainty premium needs to be added to asset pricing models. Furthermore, Chen and Epstein (2002) identified a separate premium for ambiguity, next to the risk premium.

Bali et al. (2017) measured uncertainty as the degree of disagreement between the expectations of professional forecasters (Survey of Professional Forecasters) in the economic state variables output, inflation, and unemployment. They used uncertainty betas to predict the cross-sectional dispersion in future stock returns and control for the Fama and French (1993) and Carhart (1997) factors. As a result, they found a significantly negative relation between the uncertainty beta and future stock returns. Bali et al. (2017) conclude that "macroeconomic uncertainty is a powerful determinant of cross-sectional differences in stock returns". Additionally, Chen et al. (2021) investigated the impact of economic uncertainty on stock returns in Australia. They relate economic uncertainty to lower future economic activity, which can partly be explained by the precautionary savings effect introduced by Jurado et al. (2015). A negative relationship between uncertainty and future individual stock returns was identified. However, Anderson et al. (2009) found a positive relation between stock returns and uncertainty when the latter is high, but no relationship when uncertainty is low.

### 2.4. Contribution

Concerning the impact of uncertainty in macroeconomic variables on asset returns, different methods for measuring uncertainty have been proposed and contradictory results have been found. This study can fill the gap in the literature on the impact of uncertainty in macroeconomic variables on stock returns by conducting an empirical study using a survey-based measure of uncertainty. The advantage of this method is that rather simple and intuitive measures of uncertainty are constructed. A disadvantage is that, as some researchers argue, this method does not fully capture uncertainty. Therefore, crucial simplifying assumptions are made. To summarize, this research will contribute to the

literature by providing a clear empirical study based on a Consensus Economics survey from which the cross-sectional dispersion in forecasts will be used as a proxy of uncertainty. The general effect of macroeconomic uncertainty on stock returns will be decomposed into the effect on 49 different industries, which reveals interesting patterns and insights.

## 3. Data and Variables

### 3.1. Measuring Uncertainty

A crucial assumption of this paper is that uncertainty can be fully captured using survey-based forecasts. Investors take forecasts into account in their investment decisions and thus heavily influences their behavior. These forecasts reflect where economic agents believe the economy is going. A more in depth analysis of the different ways of measuring macroeconomic uncertainty can be found in Section 2.2.

Uncertainty is measured based on a survey of Consensus Economics, compromising monthly forecasts on fifteen macroeconomic variables for over twenty countries made by a set of large banks and several companies and universities.

Each forecaster, about 20 to 35 in total, provides their prediction for the percentual change or the absolute level of the macroeconomic variable for the current and the following year. Since we will work with one-year-ahead forecast, the weighted average of the months left in the current year or forecasted months in the next year and its respective forecast is calculated. In this way, we get a forecast for the next year per economic variable for each forecaster on a monthly basis. This method is formalised in the following equation:

$$\mathbb{E}_t(z_{t+12}) = \frac{m}{12} * \mathbb{E}_t(z_\tau) + \frac{12-m}{12} * \mathbb{E}_t(z_{\tau+1}), \tag{1}$$

with (i) $m \in [1, 12]$ being the months left in the current year $\tau$, where $\tau$ is the year when the the forecast is taken, (ii) $\mathbb{E}_t$ is the consensus forecast for the economic variable $z$.

Ten macroeconomic variables were selected based on their importance in explaining stock returns (see Section 2.1). A more detailed description of the macroeconomic variables for which uncertainty will be measured is given below.

- Personal Consumption: "Measures consumer spending on goods and services in the U.S. It includes for example food, housing (rent), leisure, education etc. Also durables are included (for example cars), but not households' purchases of dwellings, which are counted as household investment;"
- Business Investment: "Measures the value of acquisitions of new or existing fixed assets with a life span of more than one year by the U.S. business sector;"
- Corporate Profits: "Represents the portion of the total income earned from current production that is accounted for by U.S. corporations;"
- Industrial Production: "Measures the output of the industrial sector, which typically comprises mining, manufacturing, utilities and, in some cases, construction;"
- Consumer Prices: "An overall increase in the Consumer Price Index (CPI), which is a weighted average of prices for different goods;"
- Producer Prices: follows the Producer Price Index (PPI), which "measures the average change over time in the selling prices received by domestic producers for their output;"
- Unemployment Rate: "Is defined as the percentage of unemployed workers in the total labour force. Workers are considered unemployed if they currently do not work, despite the fact that they are able and willing to do so. The total labour force consists of all employed and unemployed people within an economy;"
- Three Month Interest Rate: reflects the 3 Month (U.S.) Treasury Bill Rate;
- Ten Year Yield: reflects the 10 Year (U.S.) Treasury Bond Yield;
- Spread: is defined as the difference between the one year ahead forecast of the Ten Year Yield minus the one year ahead forecast of the Three Month Interest Rate.

After collecting the complete data set of one year ahead forecasts for each macroeconomic variable for each forecaster for each month, the average, standard deviation,

interquartile range and high-minus-low of these series could easily be calculated. By doing so, a time series for 360 months (1990–2019) with the uncertainty measures "standard deviation ($\sigma$)", "interquartile range (IQR)" and "high-minus-low ($\Delta$)" for each macroeconomic variable was created. A higher disagreement between the different forecasters will result in more extreme values for the uncertainty proxies used, indicating higher uncertainty.

A brief summary per uncertainty measure is given below.

- $\sigma$: per macroeconomic variable and for each month, the standard deviation of the one year ahead forecasts over all forecasters is calculated;
- IQR: per macroeconomic variable and for each month, the first quartile of the one year ahead forecasts over all forecasters is calculated and subtracted from the third quartile;
- $\Delta$: per macroeconomic variable and for each month, the minimum value of the one year ahead forecasts over all forecasters is calculated and subtracted from the maximum value.

A fourth uncertainty measure, named GARCH (Table 1), has been put in place following an AR- and GARCH-process, which is described in more detail in the methodology section.

Additionally, a second version of these uncertainty measures is created, named $\sigma 2$, IQR2, $\Delta 2$ and UNC2. They form the lagged uncertainty measures as you find the value at t-1 at time t. In this way, we can test for in-sample predictive power.

### 3.2. Stock Returns

Next, we turn to the stock returns. For this, the data library on the website of Tuck Business School in Dartmouth, New Hampshire (Kenneth R. French's Data Library) were consulted. This data library links data from the Compustat with the CRSP database.

From this data library, monthly stock returns from equities on the U.S. stock exchange were retrieved. The returns are provided for 49 different industries based on their four-digit SIC code. It concerns equally weighted returns.

Furthermore, daily stock returns were used to calculate the monthly standard deviation of the stock returns as they can serve as a measure of return volatility.

### 3.3. Control Variables

A lot of research has been conducted to determine which factors can explain asset prices and returns. First, the CAPM (Sharpe (1964) and Lintner (1965)) only saw the market beta as a relevant factor. After that, other factors were introduced to come to the Fama-French 5 factors (Fama and French (1993) and Carhart (1997)), namely:

- M = the Market Premium: $\beta_M$ captures the systematic risk, scaled by the market premium which is the market return minus the risk-free return;
- SMB = Small Minus Big: $\beta_{SMB}$ captures the size effect, scaled by the return on small firms minus the return on big firms;
- HML = High Minus Low: $\beta_{HML}$ captures the value effect, scaled by the return on high book-to-market stocks minus the return on low book-to-market stocks;
- CMA = Conservative Minus Aggressive: $\beta_{CMA}$ captures the investment factor, which is scaled by the difference in returns between firms that invest conservatively with firms that invest aggressively;
- RMW = Robust Minus Weak: $\beta_{RMW}$ captures operating profitability, scaled by the return on firm stocks with low operating profitability subtracted from firm stocks with high operating profitability.

As these factors are found to best describe stock returns, they will be used as control variables. In this way, one can test whether our uncertainty measures have additional explanatory power. Data on all the Fama–French five factors described above could also be found on the data library on the website of Tuck Business School in Dartmouth, New Hampshire (Kenneth R. French's Data Library).

**Table 1.** Summary of all variables.

| | |
|---|---|
| GARCH | Uncertainty measure based on an AR and GARCH model |
| $\sigma$ | Uncertainty measure based on the standard deviation of the one year ahead forecasts |
| IQR | Uncertainty measure based on the interquartile range of the one year ahead forecasts |
| $\Delta$ | Uncertainty measure based on the high-minus-low of one the year ahead forecasts |
| GARCH2 | Uncertainty measure based on the lagged values of the GARCH measure ( t-1 at time t) |
| $\sigma 2$ | Uncertainty measure based on the lagged values of the $\sigma$ measure (t-1 at time t) |
| *IQR*2 | Uncertainty measure based on lagged values of IQR measure (t-1 at time t) |
| $\Delta 2$ | Uncertainty measure based on the lagged values of the $\Delta$ measure (t-1 at time t) |
| M | Control variable capturing systematic risk |
| SMB | Control variable capturing the size effect |
| HML | Control variable capturing the value effect |
| CMA | Control variable capturing the investment effect |
| RMW | Control variable capturing the operating profitability effect |
| Industry3 | Monthly return on stocks in industry "Industry" |
| Industry4 | Standard deviation of monthly return on stocks in industry "Industry" |

## 4. Methodology

To measure the impact of uncertainty in macroeconomic variables on stock returns, a linear regression model was performed by means of the statistical software STATA. The research question was tested by the following general regression model for industry $i \in [1; 49]$ because of the 49 industries at time $t \in [1; 360]$ because 30 years of data were used. The implementation of this general regression model can be found from Equations (4)–(19).

$$r_t^i = \beta_0 + \beta_1 * Uncertainty\ Measure_t + Control\ Variable_t. \tag{2}$$

Next to the three uncertainty measures ($\sigma$, IQR and $\Delta$) described in the data and variables section, a fourth measure of uncertainty will be crucial in this paper. A method based on an Autoregressive (AR-) and Generalised Autoregressive Conditional Heteroskedasticity (GARCH-) model was applied, for which the procedure will be discussed below.

First, for each of the ten selected macroeconomic variables, an AR(x) regression or autoregressive model with x lags of the average one year ahead forecasts over all forecasters, was run. The number of lags used depends on the partial autocorrelation plot of the corresponding macroeconomic variable. The partial autocorrelation was considered as we only wanted to determine the direct effect of lagged values of the macroeconomic variable. We performed this at a 5% significance level so that only the most significant direct effects were used in the autoregressive model.

This procedure was repeated for each of the ten macroeconomic variables, so that the error term for each macroeconomic variable and time period ($u_{j,t}$) could be extracted. This provided a set of ten time series consisting of the error term for each time period. To capture the change in the variance of this error term over time, a GARCH model was applied.

The GARCH(p,q) model or Generalised Autoregressive Conditional Heteroscedasticity model (Bollerslev ([1986](#))) is a widely used model to estimate volatility in financial markets. It extends the ARCH(q) model by including a moving average component (p) to the autoregressive part (q). A GARCH(1,1) model was applied since it is the most used specification and should be sufficient for this research. The equation of the GARCH(1,1) model with $h_t = \sigma^2(u_{j,t}$ given $u_{j,t-1})$, $u_{j,t}$ = error term AR(x) model at time t, $t \in [1, 360]$ and $j \in [1, 10]$ goes as follows:

$$h_{j,t} = c + b * u_{j,t-1}^2 + d * h_{j,t-1}. \tag{3}$$

After executing the GARCH(1,1) regressions for each macroeconomic variable ($j$), the $h_{j,t}$ time series, which is the conditional variance of $u_{j,t}$ given $u_{j,t-1}$, was extracted and the square root was taken to obtain the GARCH-uncertainty measure. Up to now, four uncertainty measures had been constructed: standard deviation ($\sigma$), interquartile range (IQR), high-minus-low ($\Delta$) and the GARCH-measure (GARCH).

After collecting the monthly data on the industry returns and on the control variables (see data and variables section), four different effects could be tested by running four different linear regressions. All these regressions were run at a 10% significance level.

1. Impact of uncertainty in macroeconomic variables at time t on stock returns at time t
   These regressions cover the intent of this paper. The $\beta_1$ coefficient will indicate whether the uncertainty measures have a significantly positive, negative or no significant impact on the industry returns. As explained before, no clear and watertight hypothesis could be derived. With $r_t^i$ equally weighted return for industry i at time t (e.g., Agriculture3), $t \in [1, 360]$ and $j \in [1, 10]$.

$$r_t^i = \beta_0 + \beta_1 * v_t^{\sigma,j} + \beta_2 * f_t^{HML} + \beta_3 * f_t^{CMA} + \beta_4 * f_t^{SMB} + \beta_5 * f_t^{RMW} + \beta_6 * r_t^M + \epsilon_{i,t} \quad (4)$$

$$r_t^i = \beta_0 + \beta_1 * v_t^{IQR,j} + \beta_2 * f_t^{HML} + \beta_3 * CMA_t + \beta_4 * f_t^{SMB} + \beta_5 * f_t^{RMW} + \beta_6 * r_t^M + \epsilon_{i,t} \quad (5)$$

$$r_t^i = \beta_0 + \beta_1 * v_t^{\Delta,j} + \beta_2 * f_t^{HML} + \beta_3 * CMA_t + \beta_4 * f_t^{SMB} + \beta_5 * f_t^{RMW} + \beta_6 * r_t^M + \epsilon_{i,t} \quad (6)$$

$$r_t^i = \beta_0 + \beta_1 * v_t^{GARCH,j} + \beta_2 * f_t^{HML} + \beta_3 * CMA_t + \beta_4 * f_t^{SMB} + \beta_5 * f_t^{RMW} + \beta_6 * r_t^M + \epsilon_{i,t}; \quad (7)$$

2. Impact of uncertainty in macroeconomic variables at time t-1 on stock returns at time t.
   To test whether the uncertainty measures have in-sample predictive power, a regression of the returns in month t on the uncertainty measure at time t-1 was performed. With $r_t^i$ = equally weighted return for industry i at time t-1 (e.g., Agriculture3), $t \in [1, 360]$ and $j \in [1, 10]$.

$$r_t^i = \beta_0 + \beta_1 * v_t^{\sigma2,j} + \beta_2 * f_t^{HML} + \beta_3 * CMA_t + \beta_4 * f_t^{SMB} + \beta_5 * f_t^{RMW} + \beta_6 * r_t^M + \epsilon_{i,t} \quad (8)$$

$$r_t^i = \beta_0 + \beta_1 * v_t^{IQR2,j} + \beta_2 * f_t^{HML} + \beta_3 * CMA_t + \beta_4 * f_t^{SMB} + \beta_5 * f_t^{RMW} + \beta_6 * r_t^M + \epsilon_{i,t} \quad (9)$$

$$r_t^i = \beta_0 + \beta_1 * v_t^{\Delta2,j} + \beta_2 * f_t^{HML} + \beta_3 * CMA_t + \beta_4 * f_t^{SMB} + \beta_5 * f_t^{RMW} + \beta_6 * r_t^M + \epsilon_{i,t} \quad (10)$$

$$r_t^i = \beta_0 + \beta_1 * v_t^{GARCH2,j} + \beta_2 * f_t^{HML} + \beta_3 * CMA_t + \beta_4 * f_t^{SMB} + \beta_5 * f_t^{RMW} + \beta_6 * r_t^M + \epsilon_{i,t}. \quad (11)$$

3. Impact of uncertainty in macroeconomic variables at time t on standard deviation of stock returns at time t.
   Here, the standard deviation of the returns was regressed on the uncertainty measures. The former captured the volatility of stock returns. It would be expected that uncertainty in macroeconomic variables increases the volatility of the stock returns, so that $\beta_1$ turns out to be positive. With $\sigma(r_t^i)$ = the standard deviation of the equally weighted return for industry i at time t-1 (e.g., Agriculture4), $t \in [1, 360]$ and $j \in [1, 10]$.

$$\sigma(r_t^i) = \beta_0 + \beta_1 * v_t^{\sigma,j} + \beta_2 * f_t^{HML} + \beta_3 * f_t^{CMA} + \beta_4 * f_t^{SMB} + \beta_5 * f_t^{RMW} + \beta_6 * r_t^M + \epsilon_{i,t} \quad (12)$$

$$\sigma(r_t^i) = \beta_0 + \beta_1 * v_t^{IQR,j} + \beta_2 * f_t^{HML} + \beta_3 * CMA_t + \beta_4 * f_t^{SMB} + \beta_5 * f_t^{RMW} + \beta_6 * r_t^M + \epsilon_{i,t} \quad (13)$$

$$\sigma(r_t^i) = \beta_0 + \beta_1 * v_t^{\Delta,j} + \beta_2 * f_t^{HML} + \beta_3 * CMA_t + \beta_4 * f_t^{SMB} + \beta_5 * f_t^{RMW} + \beta_6 * r_t^M + \epsilon_{i,t} \quad (14)$$

$$\sigma(r_t^i) = \beta_0 + \beta_1 * v_t^{GARCH,j} + \beta_2 * f_t^{HML} + \beta_3 * CMA_t + \beta_4 * f_t^{SMB} + \beta_5 * f_t^{RMW} + \beta_6 * r_t^M + \epsilon_{i,t}. \quad (15)$$

4. Impact of uncertainty in macroeconomic variables at t-1 on standard deviation of stock returns at time t.

With this regression, the in-sample predictive power of the uncertainty measures was tested. The coefficient $\beta_1$ was also expected to be positive. With $\sigma(r_t^i) =$, the standard deviation of the equally weighted return for industry i at time t-1 (e.g., Agriculture4), $t \in [1, 360]$ and $j \in [1, 10]$.

$$\sigma(r_t^i) = \beta_0 + \beta_1 * v_t^{\sigma2,j} + \beta_2 * f_t^{HML} + \beta_3 * CMA_t + \beta_4 * f_t^{SMB} + \beta_5 * f_t^{RMW} + \beta_6 * r_t^M + \epsilon_{i,t} \quad (16)$$

$$\sigma(r_t^i) = \beta_0 + \beta_1 * v_t^{IQR2,j} + \beta_2 * f_t^{HML} + \beta_3 * CMA_t + \beta_4 * f_t^{SMB} + \beta_5 * f_t^{RMW} + \beta_6 * r_t^M + \epsilon_{i,t} \quad (17)$$

$$\sigma(r_t^i) = \beta_0 + \beta_1 * v_t^{\Delta2,j} + \beta_2 * f_t^{HML} + \beta_3 * CMA_t + \beta_4 * f_t^{SMB} + \beta_5 * f_t^{RMW} + \beta_6 * r_t^M + \epsilon_{i,t} \quad (18)$$

$$\sigma(r_t^i) = \beta_0 + \beta_1 * v_t^{GARCH2,j} + \beta_2 * f_t^{HML} + \beta_3 * CMA_t + \beta_4 * f_t^{SMB} + \beta_5 * f_t^{RMW} + \beta_6 * r_t^M + \epsilon_{i,t}. \quad (19)$$

## 5. Results

First, the general effects of uncertainty in macroeconomic variables on U.S. stock returns will be discussed. Next, we will zoom in on the separate effects per macroeconomic variable. To conclude the results section, the most important and remarkable industry effects and the connection between these different effects will be discussed.

### 5.1. General

The general analysis of our research consists of two parts. First, the impact of uncertainty on U.S. stocks will be covered. Second, we will examine implications of this impact on the standard deviation of the returns.

The main findings of the impact of macroeconomic uncertainty on stock returns are presented in Table 2. Column 1 consists of the different uncertainty measures that we have elaborated on. In column 2, we have added the number of industries that were significantly impacted when uncertainty emerged. In terms of column 2, the results are consistent, ranging from 11.4 to 15.8 out of 49 industries being significantly impacted.

**Table 2.** Overview of significance and coefficient of the impact on (standard deviation of) stock returns per uncertainty measure.

| Returns | Significant Returns | Significant SDEV Returns |
|---|---|---|
| GARCH | 12.7 | 44.6 |
| $\sigma$ | 15.2 | 41.7 |
| IQR | 11.4 | 40.7 |
| $\Delta$ | 12.7 | 39.6 |
| AVERAGE | 13 | 41.7 |
| GARCH2 | 13.2 | 43.3 |
| $\sigma2$ | 15.8 | 39.2 |
| IQR2 | 13.3 | 38.1 |
| $\Delta2$ | 13 | 36.1 |
| AVERAGE | 13.8 | 39.2 |

A key finding of our research is that, on average, there exists a positive relation between uncertainty in macroeconomic variables and stock returns. The mainly positive relation can be explained by the risk-return trade-off stated in the CAPM (Sharpe (1964) and Lintner (1965)) in combination with the uncertainty-return trade-off. As uncertainty increases, the 'uncertainty premium' (Anderson et al. (2009)) will increase, which pushes up the expected return.

The second main result concerns the impact of macroeconomic uncertainty on the standard deviation of stock returns. Although the impact on stock returns still concluded somewhat unsettled, the impact on the returns' standard deviations was unequivocally clear.

The results are presented in the third column of Table 2. We discovered a significant effect for 41.7 out of the 49 industries, with no major differences between the uncertainty measures. Hence, we can conclude that there is a very clear positive impact of macroeconomic uncertainty on the standard deviation of stock returns in the United States of America. This finding is rather intuitive since increased uncertainty mostly causes stock prices to be more volatile. As for the lagged uncertainty measures, the results are mainly the same in terms of significance. This confirms the in-sample predictive power of our uncertainty measures.

We have added a sample of our results in Figure 1, visualizing the impact of macroeconomic uncertainty in personal consumption and industrial production on the standard deviation of stock returns.

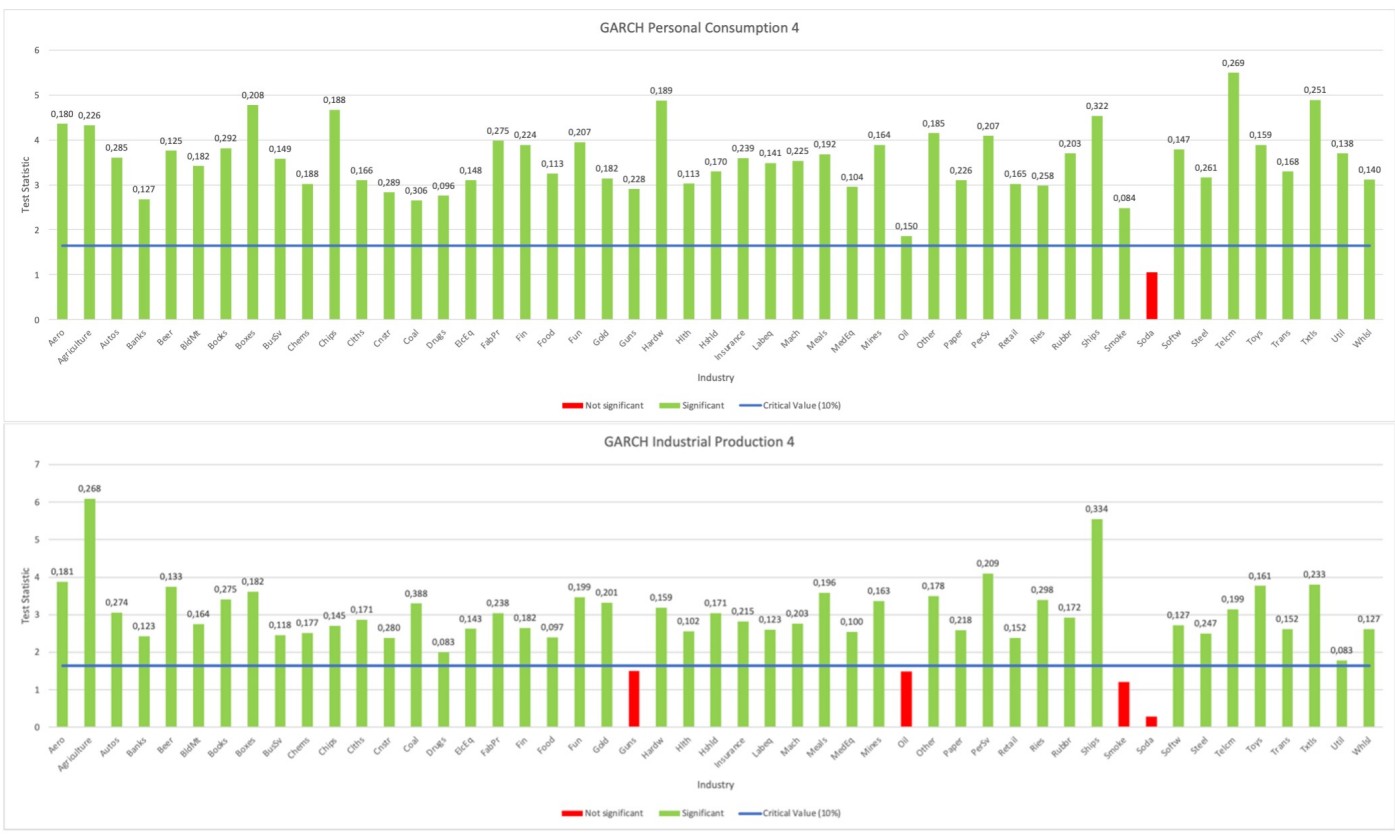

**Figure 1.** Visualisation of the impact of uncertainty in Personal Consumption and Industrial Production on the standard deviation of stock returns.

*5.2. Per Variable*

As discussed before in this research paper, we examined the effect of uncertainty in ten macroeconomic different variables on (the standard deviation of) stock returns. Per variable, we will also scrutinize how our results are related to an uncertain macroeconomic circumstances. The uncertainty in some variables might be closely related to one another because of the economic interdependency. Hence, we will examine some variables together, in an attempt to deduce logical relations in our results.

5.2.1. Industrial Production and Interest Rates

Since the financial crisis in 2007–2008, central banks kept interest rates at close to zero to foster economic growth. Since the inflationary pressures in 2022; however, interest rate increases have become ubiquitous. The increase in interest rates is an attempt of central

banks to cool off the economy, and thus by definition industrial activity. This indicates that uncertainty in industrial production is highly correlated with interest rate uncertainty, a phenomenon that becomes increasingly obvious in the post-COVID-19 period.

Uncertainty in industrial production results in the most significant impact on stock returns and the standard deviation of stock returns, with 18 to 29 sectors being impacted significantly. Considering the lagged uncertainty measures, similar impacts of uncertainty in industrial production are found. Concerning interest rates, we have pooled the impacts of uncertainty in the three-month interest rates, ten-year yield, and the spread. We have found both the lagged and the non-lagged results to be insignificant. In our opinion, this is a strange result given that the size of the spread is proven to be a solid indicator for future crises. Even for the banking sector, we have found that there mostly is an insignificant effect, with sometimes a significant negative reaction on uncertainty in the three-month yield or the spread.

### 5.2.2. Consumer Prices and Producer Prices

The difference between consumer prices and producer prices is called the mark-up of a product. When a company can charge a mark-up to customers, it has pricing power. We expected upfront that highly concentrated industries would not experience any impact of uncertainty in consumer and producer prices. For companies active in a competitive market, one would expect a significant impact of both uncertainty in producer and consumer prices. The results did not confirm our expectations. Combining all uncertainty measures, only a mere 40% of the industries were impacted significantly by uncertainty. An interesting question would be: "If a sector is impacted significantly by inflationary risk, is this the case for producer, consumer prices or both of them?" Considering the results of the GARCH measure, we found that this is very often not the case. Consequently, the question rises if this indicates potential pricing power within these sectors.

### 5.2.3. Business Investment and Corporate Profits

Since strategic investments are key to pushing a business to a new level, corporate profits should result from them. Uncertainty in business investment could result in management cutting budgets for key investments, resulting in lower future corporate profits. All uncertainty measures show significant impact for 80% of time. This illustrates a very strong relationship between the two variables.

For the effect of uncertainty on corporate profits, we found a significant effect on retail, wholesale, fun, gold, and meals for all uncertainty measures. Except for gold, these are all highly cyclical sectors. If we consider the results for three out of four measures, we can add the following industries: shipping containers, chemicals, electronic equipment, consumer goods and rubber and plastic products. These also have a very cyclical nature, which means that they are very dependent on the general macroeconomic state of the economy.

### 5.3. Sector Specific Results

Next, we will focus on the effect of macroeconomic uncertainty on the different industries. By doing so, we will make connections between the results and try to explain some of these links. The results are divided into four categories, being:

- Insignificant effect;
- Significant–Negative;
- Significant–Positive;
- Remarkable effects.

### 5.3.1. Insignificant Effect

The results showed an insignificant impact on three industries—insurance, construction, and electrical equipment industries.

In terms of insurance, one could argue that this is a logical effect since people simply do not care about the presence of macroeconomic uncertainty when considering returns in the insurance industry.

The second industry not being impacted significantly is construction. According to us, this is an unexpected result because of its highly cyclical nature. Of course, this depends on what stocks are included in this industry. Construction companies are highly dependent on how their customer base is composed (B2B compared to B2C). This divergence became ultimately clear in the financial performance of construction companies in the COVID-19 period when the B2C construction activity collapsed, while B2B construction remained on the same level.

Initially, we expected upfront of several sectors to be immune to macroeconomic uncertainty. These sectors are often described as the more defensive sectors, referring to their low beta values. These include among others food products, agriculture, and pharmaceutical products (drugs). For agriculture and food, this was the case, only excluding minor exceptions. Pharmaceutical products (drugs), on the other hand, varied notably per uncertainty measure. For the $\Delta$ and $\sigma$ measures, we found a high susceptibility for broad uncertainty, while uncertainty measured by GARCH and IQR showed much less significant results. Second, we also expected industries like tobacco products and alcoholic beverages to react mostly insignificant to uncertainty in the macroeconomic variables, given the long-lasting effects these products can exert on consumers. Looking at the results, most of the effects are indeed insignificant, in combination with two to three significant and negative results per uncertainty measure. This will be covered in the subsequent subsection.

### 5.3.2. Significant–Negative

The significant and negative results that emerged out of our STATA model consisted of a broad mix of sectors. We will further try to categorize as much of the related industries as possible.

Both the tobacco products and alcoholic beverages industries have two indicative characteristics. First, they both are classified as a pure consumer staples product, characterized by their nature of being improbable to be cut out of consumers' budgets regardless of their financial situation. This results in the second characteristic, being that they exert an addicting affect on its consumers, resulting in a non-cyclical persona. Hence, our expectations upfront estimated these two industries to react insignificantly, or slightly positively, to macroeconomic uncertainty.

Looking at the results, however, the uncertainty measure based upon the GARCH model causes uncertainty in business investment, personal consumption, and industrial production all to react in a significant and negative way. For the other three measures, results only point out a negative impact of uncertainty in unemployment rate on stock returns in both industries. This indicates that people are only changing their consumption in beer and cigarettes in a significant way if they are uncertain about future (un)employment.

In the utilities sector, we find the most negative and significant impact due to uncertainty in the ten-year yield, the spread and business investment. The latter is rather straightforward because of the cyclical nature of the industry. As discussed priorly, the amount of business investments is extremely subject to macroeconomic uncertainty. Once business investments are cut back, demand for utilities might also decrease equally.

Stock returns in the banking sector were mostly impacted in a significant and negative way by price inflation (both producer and consumer). Remarkably, uncertainty in the spread only shows a significant impact for the GARCH measure and not for the three other measures. This can be explained using a different but corresponding sector—the real estate sector—where we found negative impact for uncertainty in the three-month interest rate ($\Delta$ and $\sigma$) and the spread (GARCH).

### 5.3.3. Significant–Positive

Given the main conclusion of our research (discussed infra), it comes as no surprise that we mostly found a significant and positive impact on stock returns in the different industries. Hence, this covers the biggest group of industries that were impacted significantly by macroeconomic uncertainty. The most coherent impact was found for the following industries: clothing, food, guns, non-alcoholic beverages, and steel. The first four industries are all classified as industries with a B2C sales model. The steel industry, however, is a mere B2B type of industry.

In terms of expectations, we expected upfront that stocks in the non-alcoholic beverages, food and weapon industries would only be impacted in an insignificant way by macroeconomic uncertainty. It is logical, we thought, that consumers and governmental entities would not be affected in their consumption pattern for such non-cyclical goods. However, results showed that especially consumer and producer price inflation exercised a particularly important on the returns of these industries' stocks. Since we are reaching 'industries' as an aggregate, we do not distinguish between different subcategories and a comparison between white label products and A-branded goods are therefore out of scope for this research paper.

Wholesale and retail are two industries, the performances of which are highly dependent on inflationary pressures due to its highly cyclical nature. Consumers are inclined to cut back first on expenditures in these kinds of industries. Especially in the retail industry, we have found a positive and significant effect on the return for each of the four uncertainty measures. This is the case for both uncertainty in the consumer and producer prices. For three out of four uncertainty measures, we also found that uncertainty in business investment, corporate profits and industrial production all have a significant and positive effect on the return of both the wholesale and retail industry.

An obvious positive relationship confirmed by our results is that between uncertainty and gold. In the stock market crash due to COVID-19 in March 2020, gold prices rose to a an all-time high of 65.000 USD/kg. It is an adage that gold serves as hedge against inflationary pressures and is therefore known as an all-round safe haven during times of macroeconomic uncertainty escalation. Su (2022) found that the gold price may increase during certain periods of high macroeconomic uncertainty to hedge risks of losses, and it also shows a declining trend during periods of low macroeconomic uncertainty. However, Barro (2016) found evidence that gold may not serve at full against uncertainty risks. They discover the average annual gold return to only differ insignificantly in times of severe economic distress. Our results also show a coherent significantly positive impact on the gold industry for the basic measures. The measure based upon the GARCH method, however, showed an insignificant effect on gold returns for uncertainty in the unemployment rate and the ten-year yield. The latter seems rather counterintuitive since gold reacts negatively to a higher interest rate.

### 5.3.4. Remarkable Effects

What is interesting about the relationship between macroeconomic uncertainty and the standard deviation of the returns is that the results show coherency in terms of insignificant results for similar industries. We found standard deviations in the tobacco, drugs, oil, soda, and weapon industries to react insignificantly to macroeconomic uncertainty. The effect we have found here is quite logical, since these are mostly sectors which are not prone to a changing consumption pattern in case of economic distress. People who are sick will above all still need their medication, no matter how uncertain times are. Tobacco products (and potentially soda) are, as mentioned already above, products which cause addictive effects, so that the industry is also much less volatile in times of economic distress. However, one caveat here is that, for nearly all sectors, the effect is positive and significant for uncertainty in industrial production and inflation in producer prices. The latter could give an important indication on the general pricing power within the industry.

A general and important effect found is that two related industries very often react in the same way to uncertainty. For example, if we have a closer look at the hardware and software industries, we have found that, over all four uncertainty measures, uncertainty in nine out of ten macroeconomic variables impact industries in a similar way. For the construction materials and construction industries, similar results have been established.

For the effect of uncertainty in industrial production, exceptional results are to be noted. First, uncertainty in industrial production impacts returns in industries concerning input commodities, such as oil, coal, and utilities in an insignificant way. Also, it is remarkable that uncertainty in industrial production results in an unclear impact on the mining industry. Next, uncertainty in industrial production shows the most significant results on industries that are very consumer-related, whereas this is not shown for the more industrial sectors. Most of the positively significant effects are to be found in sectors like wholesale, consumer goods, autos, clothes, toys, software, and hardware. This is diametrically opposed to the industries on which we found an insignificant effect, being construction materials, electrical equipment, fabricated products, ships, and steel.

## 6. Conclusions

The results show a clear positive relation between uncertainty in macroeconomic variables and asset returns, which can be explained by economic theory on the risk premium. The risk–return trade-off introduced by Markowitz (1952) forms the basis of portfolio selection. However, evidence of another trade-off, the uncertainty-return trade-off is found in the literature (Anderson et al. (2009)). This could explain why the expected return, and therefore returns, increase when uncertainty increases.

The hypothesis on the effect of uncertainty in macroeconomic variables on the standard deviation of stock returns was less ambiguous. Uncertainty concerns a situation where nobody exactly knows what the future will bring. Therefore, small changes in the information set, which will remove some of the uncertainty, will have a great impact on stock markets. This sharp reaction to new information causes stock returns to be very volatile. A positive relation between uncertainty in macroeconomic variables and the standard deviation is thus expected, which is also confirmed in this research.

The results were discussed at three different levels: at a general level, at the level of the macroeconomic variables and at the sector level. At a general level, a significant impact of uncertainty in macroeconomic variables on stock returns was identified for 13 out of 49 industries on average. Regarding the impact on the standard deviation of returns, a significant effect was found for, on average, 41.7 out of 49 industries.

Zooming in on the macroeconomic variables, uncertainty in the macroeconomic variables related to interest rates came out to be predominantly insignificant. Furthermore, a strong connection was found between the results of uncertainty in business investment and corporate profits. Also uncertainty in industrial production, which has a lot of applicable industries within the data set, was found to be an important determinant in stock returns as its standard deviation.

At the sector level, many connections could be made. Considering the completely insignificant effects, the insurance industry was a logical result. However, for more defensive or addictive industries such as food products, pharmaceutical products, soda or tobacco products, significant effects were found. As for the positively significant effects, more cyclical sectors were identified such as retail, wholesale, fun, consumer goods etc. Furthermore, a positive effect was found for a countercyclical industry, namely the gold industry. This is quite intuitive since gold is a well-known hedge against uncertainty.

**Author Contributions:** Conceptualization, L.I.; methodology, R.C. and M.V.; software, R.C. and M.V.; validation, L.I., R.C. and M.V.; formal analysis, R.C. and M.V.; investigation, R.C. and M.V.; resources, L.I.; data curation, R.C. and M.V.; writing—original draft preparation, R.C. and M.V.; writing—review and editing, R.C. and M.V.; visualization, R.C. and M.V.; supervision, L.I.; project administration, R.C. and M.V.; funding acquisition, L.I. All authors have read and agreed to the published version of the manuscript.

**Funding:** This research was partly the following research grant: ARC 18/23-089.

**Institutional Review Board Statement:** Not applicable.

**Informed Consent Statement:** Not applicable.

**Data Availability Statement:** Stock returns data is available at the Kenneth French Data Library. Consensus data is available upon subscription.

**Conflicts of Interest:** The authors declare no conflict of interest.

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
