# Peer review of "The Impact of Uncertainty in Macroeconomic Variables on Stock Returns in the USA"

_jrfm, doi:10.3390/jrfm16030189_

Round 1

Reviewer 1 Report

The paper studies the effect of macroeconomic uncertainty on stock returns and the standard deviations of the returns.

Comments:

- The introduction section should be better elaborated as this topic has been well studied.

- A comprehensive review of relevant literature needs to be discussed. The references are a bit outdated.

- A competing model and method to the proposed one is the mixed-frequency approach, which should be used to compare the results.

- In practice, there are a vast set of macroeconomic variables, which need to be considered. The authors might want to use lasso for that purpose.

- Conclusions do not seem to reflect the contributions.

Author Response

Dear reviewer
We thank you for the time you took to revise our paper. 
We did our best to address your comments, in the limited time we had. 
Best regards 

Reviewer 2 Report

The research paper discusses an interesting topic, especially for current economic context and high uncertainty, reason why I consider the article worth to be revised and published in this thematic issue of the Journal.

Overall, the article needs some major revisions.

Please consider below recommendations:

1. On the section of Introduction section, would be good to have a better emphasis on the reasons why there are needed insights on the research of the link between economic uncertainty and capital market stock returns dynamics. So far, the introduction refers mainly to measures of macroeconomic uncertainty.

2. There is a lack of Literature Review section, which should highlight help the authors to built-up a robust conceptual framework and support the authors to formulate the research hypothesis and the research objective of the paper. The synthesis of the literature should be limited only to the nexus between capital markets evolution and macroeconomic output dynamics, with: (i) clear specification of major macroeconomic indicators used for such analysis; (ii) results revealed by other researchers on those links; (iii) emphasis on main critical points of economic uncertainty over the 30 years analyzed in the paper, such as the previous financial crisis and the more recent pandemic crisis with its subsequent economic negative impact on the supply chains, and overall on the economic uncertainty, that was not properly adressed by the macroeconomic decision-makers.

3. On the section of Materials and Methods:

   - it lacks a clear and robust conceptual framework of research, that highlights how each of the macroeconomic indicators used in the research paper are relevant for the dynamics of the capital market stock returns and how industry specific can generate (or not) gaps on regression model results.

   - in Table 1 there are some variables for which is not clear the justification why they are relevant or how they are calculated, such as: M, SMB, HML, CMA, RMW;

   - it is not clear the way uuncertainty measures are aggregated, resulting the standard deviation or the quartiles; for instance, the IQR results from substracting the first quartile from the third one, reffering to each year (meaning they are aggregated opinions of different forecasters) or how? is the same measure aggregated per year, but also for different industries...?

   - maybe a graphical representation of the calendar considered for calculating the weighted economic consensus;

   - for me is not clear which are the control variables; are they the industry-based dummy variables, or the undefined variables from Table 1, which I already reffered in the previous comment?

   - is there basis of theory that justify to use all the 10 macroeconomic indicators to check if uncertainty on their estimation lead to changes on the stock returns?

   - the stock returns at time t are also some aggregate measures weighted by the proportion of the capitalization on the capital market index considered for the analysis? Eventually, I would except each regression to have monthly values forecasted for macroeconomic indicators and aggregated monthly stock returns, either based on industry or with overall aggregation; but this is not clear;

4. On the section of results:

    - there are plenty of simple regression models estimated (overall regression model, industry-based specific model, macroeconomic output-based model), which leads to lots of statistical results; however, the way the results are presented gives me no change to make a connection between figures, tables,  regression models, and interpretations; the way results are presented provides no change to follow the interpretation together with the figures from the different tables or figures, or to link them to different regression model; there is also not mentioned the t-Statistic and no significance level (would I understand for all coefficients is 5%?); the naming "average coefficient" is not usual on research papers;

   - authors talk about correlations between different variables, but the table is missing from the manuscript;

   - the presentation of the results is not referred to similar studies in the literature; the results are simply presented and there are too little spots in the paper where an economic interpretation with related implications, is given as well; this reference to the literature is even more important in case of results that even the authors underline are ambiguous, odd or remarkable; this way, the results could be positioned towards a direction of research already noted in the literature.

    - the results in the subsections Per variable, or Per sector specific are not reffered to any table with regression results; it is not clear which regression model has been used to interpret results on those sections, using which uncertainty measure;

   - Figure 1 is just mentioned in the results section, without any interpretation of results, reason why it is not clear the value add on achieving paper objectives;

  5. There is no separate Discussion section, and too less discussion on the economic implications of the paper on the Results section.

6. Most of the references are too old.

Author Response

(The authors gave the same response as above.)

Round 2

Reviewer 1 Report

The manuscript has been revised and I have no further comments.

Author Response

Dear reviewer, 
Many thanks for your inputs. 
Best regards 

Reviewer 2 Report

Most of the concerns from my side are adressed and significant increase on paper's readability is ensured. 

Author Response

(The authors gave the same response as above.)
